# TASKBASED ENGINEERING

**Abstract.** The article describes the main methodological principles of a new scientific-engineering discipline — taskbased engineering, which represents a synergistic integration of the taskbased approach, engineering and agentbased AI. The features of each component of taskbased engineering are examined,including their advantages and disadvantages. A conceptual model of the new discipline ispres ented, and its key principles, methodological framework and application potential are elucidated. Differec es from related fields are highlighted, and the challenges and development prospects are outlined.

**Keywords:** taskbased engineering, weakly structured need, taskbased approach, domain semantic mode l, solution criterion, contextual conditions, task formalization, semantic modelling, engineering, prompt en gineering, context engineering, agentbased AI, AI agents, multiagent systems.

The foundation of any intellectual activity is the ability to formulate and solve problems. However, a serious obstacle in this area stems from two key factors: on the one hand, the lack of a discipline that explains and facilitates an effective transition from an initial weakly structured need to a clearly formulated task; and on the other hand, the complex ity and multicomponent nature of the resulting welldefined problems — including their hierarchical structure, multicriteria nature, dynamic environment, data scarcity, the n eed to account for heterogeneous factors, etc.

Traditional methods for problem formulation and solving often prove insufficient due to:

- vagueness in the initial need's formulation;
- the nontrivial nature of the process of transitioning from a weakly structured probl em (expressed as a need) to a precise task formulation;
- neglect of contextual dependencies;
- linear planning of actions for transitioning from a weakly structured problem to a welldefined task and subsequently solving it, without adaptation to possible changes in conditions.

Naturally, this gives rise to the need for a methodology, theory, and technology that wou ld take the above factors into account and help overcome these shortcomings.

In the 1980s, a methodology called the **task-based approach** [1–8] was proposed and has continued to develop actively, along with its logical-mathematical theory — **semantic modelling** [9–24, 39–45]. According to the task-based approach, when discussing a task, we are required to present its following components: a semantic model of the domain to which the task belongs; query(ies) directed to the semantic model, the answer to which constitutes the task's solution; a solution criterion — i.e., whether what is presented as the answer to the query actually qualifies as a solution; contextual conditions. Practice has confirmed the promise and effectiveness of the task-based approach and its mathematical theory — semantic modelling. However, it is worth noting that the methods of the task-based approach and the algorithms of semantic modelling developed to date pay little attention to the process of transforming a weakly structured problem into a well-defined task. Furthermore, the language and algorithms of semantic modelling are

primarily oriented towards static task formulation and the problem-solving process, failing to provide adaptation of task formulation and action planning to changing conditions.

During the same period, there was a rapid emergence and development of the methodology, theory, methods, algorithms, and systems of **machine learning** — in particular, **large language models** and **multimodal models** (LLMs, LMMs). The advent of these models marked new, unprecedented opportunities in the computer "understanding" of natural-language queries and in answer generation. Nevertheless, AI systems based on neural networks often exhibit unpredictable behaviour with respect to specified requirements and context, vulnerability to attacks via subtle distortions of input data, an inability to consistently account for context, a combination of risks and opportunities, and limited capacity to effectively leverage expert knowledge. When using modern generative AI models, mechanisms for interacting with them become particularly important. One of the key interaction mechanisms is the so-called **prompt engineering** (see, for example, Google's excellent guide to prompt engineering [26]). It quickly became clear that the performance and effectiveness of generative models are largely determined not only by the adequacy of query formulation but also by the context they receive. Over time, this context — ranging from simple instructions to guidance on using complex external knowledge bases — gradually became another primary mechanism for controlling the behaviour of a generative model, expanding its knowledge, and unlocking its capabilities. Two formal disciplines have gradually emerged: **prompt engineering** and **context engineering** [26–29]. From the author's perspective, these two disciplines — which mutually complement and enrich each other — should be considered as a single discipline: **engineering**. This discipline, like the task-based approach, is oriented towards automating the processes of problem formulation and solution search. However, unlike the task-based approach, this engineering can only act as a passive toolkit.

In addition to generative artificial intelligence, another traditional area in AI has recently become extremely popular again: **agent-based AI** — a field concerned with the creation of **AI-agents** and **multi-agent systems** (MAS) (here, we can recommend the excellent three-volume monograph [25]). Today, many generative AI-models, as well as a number of specialized frameworks and platforms, enable the automatic creation of agents with various roles and competencies. They also provide the ability to integrate AI-agents into specialized communities — i.e., multi-agent systems. It should be noted that modern agent-based AI distinguishes three main directions: **Software agents** (**program agents**), particularly **LMM-agents**, which operate in the digital world, **Embodied agents**, including **cyber-physical agents**, which possess "bodies" and interact with the real world, **Hybrid agents**, which represent a combination of software and embodied agents.

Since all the aforementioned areas are, to varying degrees, oriented towards problem s olving, an analysis of their current state and capabilities naturally led the author to the d esire to unify all three disciplines. Their synergistic integration constitutes **taskbased e ngineering**.In fact, this article discusses the emergence of a new scientific and enginee ring discipline that unites:

1. **The taskbased approach** —
the *core* of taskbased engineering, its systemforming
framework, which sets the direction for transforming a weakly structured problem into
 a welldefined task and subsequently solving it.
2. **Engineering**, represented by:

o **Prompt engineering** — a *tool* that enables dialogue with a generative AI-model (e.g.,
an LLM/LMM) for efficient information retrieval, content generation, and refinement of task components;

o **Context engineering** — a *tool* through which one can create a "working environment" for the needsatisfaction process by managing its surroundings. In the case of using LLM/LMMs, this includes the system prompt, metainstructions, r etrieval of relevant data, dialogue history, etc.

3. **Agentbased AI**, including **AI-agents** and **multiagent systems (MAS)**, which en- ables the organization of an effective process for transitioning from a weakly struc- tured problem to a welldefined task, and for subsequently solving complex, multico- mponent problems.

As noted above, the core of this new discipline is the **task-based approach**, which addresses the following questions: what a task is as an entity, where and how it emerges, how it should be formulated and solved, what types of tasks exist, etc. It should be noted that it was Academician A.N.Kolmogorov who first spoke of tasks as entities deserving separate attention and thorough study — in his early work [30], published back in 1932. There, while describing the denotational semantics of intuitionistic propositional calculus in terms of tasks, he effectively laid out the **axiomatics of task calculus** for the first time. Interestingly, at the time this article was written, it was still unclear what should be understood by a "general method for solving" tasks of a certain class, and what it means to "reduce the solution of one task to the solution of another" [31].

The formal study of tasks, initiated in the 1930s by A.N.Kolmogorov, was independently continued by researchers in other disciplines. Among them, in the author's view, a notable place is occupied by the **Theory of Inventive Problem Solving (TRIZ)** [32–35], proposed in the 1950s by the Soviet patent engineer G. Altshuller. Here, the task-based approach was formulated as a purely empirical discipline, primarily oriented towards the practical, engineering application of its principles. According to TRIZ, a **task** is essentially nothing more than the need to overcome a contradiction between the desired outcome and the existing capabilities of the Problem Solver at each stage of the synthesis/analysis of a Technical System (TS).

The task-based approach received further significant development in the works of Academician Yu.L.Ershov and D.Sci. (Phil.) K.F.Samokhvalov, which are devoted to applying the ideas and principles of the task-based approach to the foundations of mathematics [36, 37]. A key point of the proposed approach is the conclusion that knowing a need "…does not mean knowing the desired outcome, but only means having the ability to recognize the desired outcome" — or, in other words, possessing a **criterion for need satisfaction.** Thus, a task — as a certain need — is defined (conceptualized) if and only if there exists a **solution criterion**: a means of verifying whether the presented construct actually constitutes a solution to the task. Otherwise, any proposed consideration could be considered a solution to the task.

The propositions formulated and substantiated in the works of Yu.L.Ershov and K.F. Samokhvalov were subsequently developed by a group of mathematicians in the context of artificial intelligence (AI) [1–8]. The need to practically apply the task-based approach for automated problem solving in AI necessitated a more detailed examination of the very concept of a "task". This examination resulted in formulating the concept first at the

methodological level and then at the formal, mathematical level. This led to the creation of an original logical-probabilistic theory of the task-based approach — **semantic modelling** [9–24, 39–45]. According to the task-based approach, a task is considered defined if and only if its formulation includes the following components:

- **Specification of the domain**, captured in the form of a model, including a description of the signature and structure of a domain-specific language (ontology), as well as general knowledge and domain-specific knowledge: initial data, facts/precedents, rules, constraints, and hypotheses [46];
- **The query (question)** that the task poses to the domain model — i.e., the specific question to which we must obtain an answer (the task's solution);
- **A criterion for query satisfaction** — specifying under what conditions we can consider that an answer (solution) to the query (question) has indeed been obtained;
- **The context** in which the answer (solution) to the query (question) should be sought and interpreted — namely: what goal we pursue by solving the task (i.e., what we expect from the obtained result and what its consequences are); what to do if the answer turns out to be negative; other conditions surrounding the task — resources, constraints, norms, etc.

Based on the specified structure of the concept of a task within the task-based approach, the following scheme of actions for need satisfaction was proposed:

**BEGINNING OF THE SCHEME**

**STEP 1**. Identification and analysis of needs.

An expert and domain analyst studies, clarifies, and conceptualizes the emerging weakly structured need, using the possibility to apply one or another existing action template that would allow satisfying it. If no suitable "template-based" method for satisfying the need is available, the associated contradiction between the desired and the actual state of affairs is identified and studied. This contradiction constitutes the true cause and content of the task to be solved.

**STEP 2.** Planning.

A plan of action is developed, including: creation of a model of the domain to which the task belongs; description of the query (queries) requiring an answer(s); formulation of a criterion for successfully overcoming the identified contradiction; analysis of the conditions for organizing the task-solving process, treating the results obtained at this stage as the initial version of the task's context.

**STEP 3.** Formation of a vision (concept) for the task formulation and solution project.

In natural language, general and specific evaluative knowledge relevant to the stated task(s) is described. Relevant facts and other precedents are collected, and an ontological model of the problem domain is constructed (including concepts, relations, properties, etc.). A class of possible queries to the problem domain is formulated in general and ontological terms — these queries reveal the content of the overall task and its subtasks. A version of the task solution criterion (for both the main task and subtasks)

is also formulated. The part of the task representation and domain understanding that requires either: external calculators acting as oracles, or "basic" entities already obtained or acquired manually during the project execution (which, incidentally, can also be treated as oracles) is identified.

**STEP 4.** Formalization of a semantic domain-specific language (sDSL) for the domain and task formulation in terms of this language.

A formal model of the problem domain is constructed, and the query and task solution criterion are formulated in the same formal terms — taking contextual conditions into account. The identified set of missing oracles is programmed, and communication with existing oracles is addressed. If a solution criterion based on testing with semantic test cases is chosen, their formal specifications are written.

Note that if STEP 4 is carried out within and using the tools of semantic modelling, we can immediately obtain a computer-based version of a system oriented towards task solving (no-code).

**STEP 5.** Testing and debugging.

**STEP 6.** Operation (deployment and use).

**END OF THE SCHEME**

It should be noted that the formal refinement of the concept of a task within the task-based approach required the creation of an original semantic model of computability. Its mathematical foundation is the concept of formula definability on constructive models [38], using oracle (relative) computability. As a formal language for task specification, a sufficiently expressive fragment of the predicate calculus language was chosen — the class of $\Delta_0$-formulas and $\Delta_0$-terms, incorporating oracle computability. This allowed restricting the scope to the class of polynomial-complexity computations without losing the expressive power of the language or its potential completeness. It has been shown that the $\Delta_0$-fragment of a many-sorted first-order predicate calculus language corresponds to the concept of polynomial computability — in other words, polynomial computability is $\Delta_0$-formula definable [17, 19–23]. Moreover, such a language of executable specifications can be effectively enriched with probabilistic constructions, which significantly expands the expressive capabilities of the declarative specification language [40–45].

Now, a few words about the second component of task-based engineering — **engineering**, which combines two mechanisms for interacting with generative AI models: prompt engineering and context engineering. Currently, **prompt engineering** is a discipline at the intersection of psychology, linguistics, and computer science. It studies methods for developing and optimizing queries (prompts) to ensure effective interaction with generative AI models — particularly with large language models (LLMs) — when solving various tasks. This interaction mechanism enables guiding the generative model's work toward producing accurate, relevant, and useful results through carefully crafted query formulation. Let us recall that generative models generate text by predicting the most probable next tokens (words or symbols) based on the input prompt and the data they were trained on. Therefore, the goal of prompt engineering is to control these

probabilities by setting the context, constraints, and reasoning direction of the generative model. The structured nature of a prompt is especially important for its effectiveness. This means the need to specify the query processing context — for example, a role, clear instructions, examples, an explicit indication of the required answer, etc. Moreover, creating a high-quality prompt often follows an iterative process. This implies conducting a certain cycle of interactions with the generative model: formulating, testing, analyzing the results, and optimizing. It is extremely useful to compile and use prompt libraries — collections of proven queries grouped by topics or styles.

To date, a wide range of techniques has been developed to improve the quality of generative model responses. Here are some of them:

- **Zero-shot prompting** — the model provides a response without using any examples; it is applied to solve simple tasks or tasks that are well-known to the model.
- **Few-shot prompting** — several task execution examples are included in the prompt, which helps the model better understand the requirements.
- **Chain-of-Thought (CoT)** — instructing the model to generate step-by-step reasoning before providing the final answer; this technique is especially effective for logical, mathematical, and multi-step tasks.
- **Self-Consistency** — the generative model generates multiple reasoning chains and selects the most frequently occurring answer, thereby increasing the reliability of the result.
- **Tree-of-Thoughts (ToT)** — the model explores several parallel reasoning paths simultaneously, which is useful for complex tasks.
- **ReAct (Reason and Act)** — the model alternates between reasoning steps and using external tools (information retrieval, code execution, etc.).
- **RAG (Retrieval-Augmented Generation)** — linking responses to factual data retrieved from external sources, which helps reduce the likelihood of errors.
- **Metaprompting** — the initial query generates a more detailed sub-query, allowing the model to refine the task.
- **Role-based prompting** — assigning a specific role to the generative model (e.g., expert, editor, guide, etc.), which helps tailor the style and content of the response.
- **Step-back prompting** — the generative model first considers a general question related to the task and then uses the acquired knowledge to solve the specific problem. This can mitigate bias and improve response quality.

Currently, prompt engineering as a discipline continues to actively evolve. Among the current trends is the development of **multimodal prompts**, which combine text with images, audio, and video.

As for **context engineering**, this branch of engineering is responsible for the systematic work with the context of interaction between the user and the generative model — in order to improve the quality, relevance, and security of the generated responses. Unlike prompt engineering (which focuses on query formulation), context engineering manages the *query's environment*: dialogue history, metadata, external sources, constraints, and other elements. As in the case of prompt engineering, context engineering has also accumulated a substantial set of techniques by now. The main ones include:

- **Dialogue history management**;
- **Thematic segmentation;**
- **Role labelling (user/assistant) for clear attribution;**
- **External context retrieval (RAG);**
- **Hybrid search (keywords + semantics) to improve accuracy;**
- **Re-ranking of results before feeding them into the generative model;**
- **Context structuring;**
- **Reliability control;**
- **Context personalization;**
- **Context constraints;**
- **Dynamic context management;**
- **Context visualization;**
- **Meta-context** in the form of explanations provided to the model about the context;
- **Context automation** using preprocessing, postprocessing, and context agents.

The range of tools and technologies used in context engineering is also extensive, including various **vector databases, RAG frameworks, context management systems, and annotation and quality monitoring tools**. In general, the role of context engineering lies in transforming "raw" data into structured knowledge. It is important to note that the optimality of a context engineering strategy is ensured by the iterative nature of interaction with the model and the ability to adapt it to a specific task.

It should also be emphasized that prompt engineering and context engineering do not compete but complement each other, forming a unified control loop for interacting with generative models. Their relationship can be figuratively described as **"form through content"**, where: prompt engineering provides the form of interaction (how to ask); context engineering provides the content — what information to feed the model, where to retrieve it from, how to structure and update the data. Thus, the prompt acts as a kind of "command line", while the context represents the data needed to obtain an accurate response. To date, several mechanisms for integrating prompts and context have been developed. The most popular ones include:

- **Explicit inclusion of context in the prompt**, where the context simply becomes part of the input query;
- **Dynamic context loading via RAG** (Retrieval-Augmented Generation), where the prompt is formulated as a special query. The system automatically retrieves the context, and it is embedded into the final prompt before it is sent to the generative model;
- **Multilevel management**;
- **Iterative optimization**, where the prompt initially sets the query, the context module loads relevant data into it, the generative model generates a response, and this response is then analyzed for quality. If needed, the prompt or context is adjusted accordingly;
- **Role separation in multi-agent systems**.

As for **agent-based AI**, research in this field began almost from the very inception of AI and was closely linked to cybernetics, automata theory, and other scientific disciplines that model the behaviour of artificial and biological entities in an external environment. In

fact, the topic of agents and multi-agent systems (MAS) almost immediately split into two directions: the area related to the creation of **software agents** and communities built from them, operating in the digital world and the area of **embodied agents** and their communities functioning in the real, physical world.

Regarding the first direction, the original idea of an intelligent software agent emerged as an attempt to intellectualize the user interface. Instead of requiring the user to initiate actions via commands, it was proposed that a computer intermediary should collaborate with the user in solving tasks. Gradually, this concept expanded beyond the scope of interfaces and began to focus on: artificial intelligence methods; the use of computer networks; distributed databases; distributed computing. In the 1980s, autonomous mobile robots capable of acting without central control were also actively developed. These robots responded to environmental changes rather than to pre-defined commands. In the 1990s, the so-called **agent-oriented approach** was finally established in AI. Within this framework, attempts were made to describe a social perspective on the organization of computation — one based on the interaction of software agents during the computational process.

With the development of machine learning and deep learning at the beginning of the current century, the concept of an intelligent software agent acquired new meaning. This is because next-generation language models have learned to maintain context, reason, and interpret queries. Starting in 2023, LLMs (Large Language Models) began to be used not as a tool, but as an element of a reasoning software agent capable of independently setting goals, formulating subtasks, and executing them. This gave rise to the concept of an **AI-agent** — an autonomous system able to: understand complex goals; independently plan the steps required to achieve them; actively interact with the surrounding digital environment. It should be noted that AI-agents differ from chatbots and intelligent assistants/helpers/co-pilots in that they can not only respond to queries, but also perform actions using various tools (APIs, databases, external services). Modern AI agents are built from modules, with the main components being: a reasoning core ("brain") based on an LLM (Large Language Model), LMM (Large Multimodal Model), or VLM (Vision-Language Model); a prompt instruction; memory with context; tools for acting in the external digital world.

AI agents are now being used to build so-called **multi-agent systems** (MAS) — communities in which several AI-agents interact with each other to solve tasks that are too complex for a single agent. The interaction can be: cooperative — for example, a team of bots collaboratively writing a report; competitive — for example, when agents negotiate the price of a delivery or compete in finding vulnerabilities. Recently, a direction has been actively developing that involves creating and using small language models (SLMs) instead of large universal models. This is associated with the development of specialized AI agents for specific tasks and domains. This has enabled the creation and development of a wide range of industry-specific solutions: AI agents specialized in finance, retail, healthcare, logistics, and so on. Various AI agent ecosystems are now being actively created, including **frameworks** and **platforms** for building and managing multi-agent systems. Generative AI models are increasingly being used as the cognitive core of AI-agents. Significant efforts are now being made to develop and implement AI safety standards, including the concept of trust, risk, and security management for AI agents and multi-agent systems. This focus on trust and security is due to the fact that

generative models are prone to generating «hallucinations». AI-agents may also: perform unauthorized actions; contribute to data leaks; participate in cyberattacks; make unexpected decisions; act in ways that are difficult to anticipate in advance — especially in dynamically changing conditions. At the same time, it should be taken into account that legal responsibility still lies with a human or an organization.

Regarding multi-agent systems (MAS), they also come with their own set of challenges. For example, there are problems related to the complexity of integrating AI agents with various information systems and corporate processes. This requires complex access and control configurations. Moreover, as the number of interacting agents increases, the complexity of coordinating their actions grows exponentially. The topic of multi-agent systems has recently gained particular relevance, as it is expected that in the future, MAS will become even more widespread and will tackle increasingly complex tasks. It is anticipated that: **hyperpersonalization** will develop, based on analysis of the user's or customer's emotional state; **proactive service** will become ubiquitous — where assistance is provided before a problem arises. In the long term, the emergence of **personal AI-agents** — digital secretaries — is possible. Such agents would: manage schedules; handle correspondence; filter information flows. However, this will require addressing ethical, legal, and technical challenges related to: control; responsibility; trust; security.

Previously, the topic of software agents and multi-agent systems (MAS) composed of them was discussed. Equally relevant is the topic of **embodied and cyber-physical AI-agents**, as well as the corresponding multi-agent systems. Hereafter, by an embodied agent we will mean AI systems that interact with the physical world through their sensors and actuators. This enables them to perceive the environment using video cameras, microphones, sensors, and other devices; make decisions based on the data received; and perform actions in the real world. The architecture of an embodied agent typically includes the following modules:

- **Perception module**, responsible for image processing (CV — Computer Vision), audio analysis (ASR — Automatic Speech Recognition), and interpretation of data from other sensors (temperature, pressure, distance, power supply, etc.);
- **Understanding module**, which handles tasks such as: building an environment map (SLAM — Simultaneous Localization and Mapping); identifying objects and events; predicting changes;
- **Reasoning module**, which performs: action planning (e.g., pathfinding, optimization); decision-making under uncertainty; learning (RL — Reinforcement Learning, simulation-based learning);
- **Action module**, including: motor/actuator control; communication with other agents; adaptation to environmental changes;
- **Memory and Learning module**, responsible for: storing experience (trajectories, successful strategies); updating models through feedback.

A subclass of embodied agents are **cyber-physical agents** — embodied agents integrated into cyber-physical systems (CPS) that combine: computational components (algorithms, models); physical elements (sensors, mechanisms); network connections (IoT, 4G/5G). It is easy to see that embodied agents differ from software agents primarily in that they interact directly with the physical environment and therefore must account for

the laws of physics (friction, gravity, inertia, temperature, etc.). Moreover, they often need the ability to process data in real time. Finally, they are typically subject to heightened requirements for reliability and security.

Multi-agent systems (MAS) with embodied agents have several distinctive features compared to software-based MAS, due to the fact that they operate in the physical world. For example, MAS agents must: "understand" and account for their physical constraints (battery life, mechanism wear, weather conditions); be aware of each other's locations and of obstacles; synchronize timing when coordinating actions; account for potential communication limitations, instability, and data transmission delays.

Multi-agent systems (MAS), both software-based and embodied, are characterized by various types of organizational interaction architecture:

- **Centralized,** where the system designates an agent that acts as a single coordinator (the term "orchestrator" has become popular recently), assigning tasks to the other agents in the system.
- **Decentralized**, where agents negotiate directly with each other. This architecture is recommended for dynamic operating environments.
- **Hybrid**, which combines features of the two previous types of agent community organization. This is usually achieved by enabling the creation of local groups (clusters) with their own coordinators, along with the presence of a global supervisor/orchestrator.

In recent years, the **role-based organizational** architecture of MAS has become very popular. In this approach, agents are pre-assigned specific roles and competencies. Below is one possible variant of a step-by-step scheme for creating a software agent with specified competencies.

**BEGINNING OF THE SCHEME**

Step 1. Define the role, goals, and functions of the software agent.
**Example:**
*Role:* Financial analyst.
*Goal/Task*: Assess the profitability of a business plan.
*Functions*: Calculate NPV, IRR, payback period; perform sensitivity analysis for interest rate changes.
Step 2. Design the agent's competencies, classifying them as basic, domain-specific, or cognitive.
Step 3. Select the technological base. This involves choosing either: a universal option — typically referring to LLM agents; specialized models; hybrid systems (LLM + scripts).
Step 4. Training and configuration. The main methods for designing the software agent here are prompt engineering and context engineering.
Step 5. Integrate the AI agent into a multi-agent system (MAS). At this stage, the interaction mechanisms among the agent community members must first be defined. This is largely determined by: the class of tasks the community solves; its organizational architecture; the role composition of the agents.

When integrating a previously created AI-agent, it is usually assumed that the MAS already includes a special agent called a "**coordinator**" or "**orchestrator**", responsible

for task decomposition and subtask distribution among agents. Typically, a data bus is used for task distribution.

**Step 6**. Testing and validation. At this stage, a specific testing scenario is selected and implemented — for both individual agents and the entire MAS. These may include, for example: **functional testing; load testing; stress testing**.

**Step 7.** Deployment and monitoring. This is one of the most labour-intensive and critical steps. For this reason, it is recommended to use ready-made tools that help implement and configure: orchestration; containerization; monitoring; logging.

**Step 8.** Iterative improvement. This step is a cyclical procedure aimed at enhancing the performance quality of both the agents and the MAS as a whole.

## END OF THE SCHEME

Previously, we mentioned the special significance and importance of the orchestrator agent in a multi-agent system (MAS). This agent is typically assigned the following functions:

- Accepting and decomposing the initial task;
- Distributing the subtasks (obtained through decomposition) among the system's agents;
- Appointing a leader and support agents for a specific scenario or cluster in a hybrid MAS architecture;
- Managing communication channels. In particular, the orchestrator is responsible for configuring the message bus or API gateways through which agents exchange events. This may involve either: a centralized task queue; a distributed list where each agent subscribes only to the topics relevant to it;
- Tracking changes in external conditions (new data, timeouts, errors from individual agents) and adjusting priorities accordingly — determining which agent should respond first, and which responses can be collected in a "batch" later;
- Monitoring and redistributing resources to ensure the system remains responsive and reliable;
- Collective verification through cross-checking schemes: validating the main solution using a verifier agent and an arbiter agent; if discrepancies with predefined criteria are detected, activating fallback mechanisms — e.g., triggering "Plan B": redirecting the task to a backup agent or initiating retraining;
- Monitoring and reporting, which involves: collecting performance metrics for each agent (response time, accuracy, load); visualizing cluster status; automatically generating notifications about critical events, system changes, or when key indicators approach threshold values.

Thanks to orchestration, a multi-agent system can operate as a single "living" organism — where agents do not merely execute their functions sequentially, but respond synchronously and adaptively to system challenges. When designing an orchestrator, it is first necessary to clearly define: what class of tasks the MAS (and thus the orchestrator) will solve; which agents will participate in the system; what interaction scenarios with the user and between agents need to be supported; what organizational architecture of the MAS is selected. Once decisions are made regarding the task class, architecture type, and role composition of the MAS, the following are defined: decision-making algorithms; rules for task decomposition and distribution; error handling mechanisms; adaptation

mechanisms for changing conditions. Here, one can use a logical-mathematical approach to build the "brain" of the orchestrator agent, or rely on a specific LLM (Large Language Model). A hybrid approach can also be applied, combining a logical model with a generative model. To date, a wide range of tools has been developed for creating the "brain" of orchestrators.

Previously, we defined task engineering as a synergistic integration of the task-based approach, engineering, and agent-based AI. In this framework, the task-based approach occupies a central place as the core of a new scientific and engineering discipline, while the other components serve as tools. Let us recall that the main problems that task engineering aims to address are:

a) Automating the transition from an unstructured need to a clear formulation of the corresponding task;

b) Automating the task-solving process itself — especially when dealing with complex, large-scale, and multi-component tasks. Solving such tasks may require: preliminary **decomposition** into subtasks; **planning actions** to address them; aggregating these partial solutions into a final solution for the entire task; accounting for necessary resources (computational, temporal, intellectual, etc.).

In addressing these fundamental problems, the **task-based approach** provides the methodological foundation for action. It specifies the direction for transitioning from an unstructured need to a structured task, and: establishes a conceptual system in which the question (requiring an answer) is formulated; fixes the task completion criterion (including quantitative and qualitative metrics); defines the logic for result verification; ensures process reproducibility. Thus, the task-based approach defines the overall **goal-oriented direction** of the entire process of satisfying the original unstructured need. The other components of task engineering serve as a "bridge" or "interface" between humans and AI systems — acting as sophisticated, yet merely instrumental tools. **Prompt engineering** enables one to clearly pose and even formalize a query to generative models, specifying the structure and format of the output. It can also guide the cognitive processes of a generative model — for example, by using Chain-of-Thought or few-shot learning. **Context engineering** ensures data relevance through RAG (Retrieval-Augmented Generation). It: updates the generative model's knowledge without retraining; filters information based on reliability and freshness criteria; creates a system "memory" for sequential task solving. Its key function is to ensure precise, complete, and up-to-date interaction with generative models. **AI-agents** and **multi-agent systems** (MAS) are among the most effective modern tools for problem solving. AI-agents, with their ability to adapt to changes via a cybernetic loop (feedback), can: perform a wide range of autonomous actions (data analysis, calculations, content generation); be highly specialized in specific domains (finance, business, science, manufacturing, construction, marketing, law, etc.). As for multi-agent systems, they are ideal for solving complex and large-scale tasks because they can: scale up; support parallel and hierarchical processing of subtasks; coordinate agent interactions (orchestration); implement verification and consensus mechanisms. Overall, AI-agents and MAS can play a key role in transforming a weakly structured problem into a formalized task and in practically solving it with quality control. The direction and content of these actions are determined by the task-based approach.

Naturally, the question arises: what exactly is the synergistic effect of integrating the components of task engineering? To answer this question, let us consider one possible variant of a **complete 4-phase lifecycle for satisfying an initial need using task engineering.**

**BEGINNING OF THE CYCLE**

**Phase 1****.** Problem formulation and conversion into a task (task-based approach + engineering with its prompt techniques and templates for structuring + agent-based AI). If necessary: decompose the task; build a dependency graph of subtasks; assign AI-agents to subtasks (task-based approach using a graph visualizer + context engineering with external data (RAG, knowledge bases, APIs) + multi-agent system concept).

***Input:*** unstructured problem description.

***Output***: formalized task statement, decomposed into subtasks + hierarchical subtask plan for agents with dependencies.

**Phase 2.** Planning with smart contract generation (engineering + agent-based AI).

***Input:*** decomposed subtask plan with dependencies assigned to MAS members.

***Output***: executable action plan with resources and deadlines.

**Phase 3**. Execution and monitoring (task-based approach + context engineering + multi-agent system concept).

***Input***: action plan for solving subtasks.

***Output:*** intermediate results + progress report.

**Phase 4**. Adaptation and lesson capture in the knowledge base (task-based approach + prompt engineering + context engineering).

***Input***: monitoring data.

***Output***: updated plan/formulation → **Phase 3.**

**END OF THE CYCLE**

Naturally, each phase of this general scheme is actually a multi-step procedure that refines the sequence and content of actions for its implementation. Below is an example of a step-by-step process that reveals the content of **Phase 1.** It is assumed that by the time this step-by-step scheme is used, the basic composition of agents in the instrumental multi-system has already been formed. This composition is generally suitable for a fairly wide range of typical needs and medium-complexity tasks with clear boundaries. It usually includes:

- **Interviewer agent** — clarifies the initial need and requests;
- **Modeling agent** — builds a semantic model of the domain to which the need and formulated task belong;
- **Formulator agent** — generates the query for which a solution (task answer) is sought;
- **Verifier agent** — defines the task completion criterion;
- **Context aggregator** — collects data;

- **Coordinator/Orchestrator** — the main agent that manages the entire lifecycle of need satisfaction. It: initiates the activity of the appropriate agent at the right moment; if necessary, decomposes a complex task and assigns subtasks to the relevant agents; integrates solution results and other components; monitors processes;
- **Tester agent** — verifies the task before execution.

However, if we encounter a complex multi-component task, then there arises a need to expand the basic composition of the multi-agent system. For example, when launching a project to create a **new medical diagnostic service**, the following additional agents will be required:

- **Medical expert** — validates algorithms against clinical guidelines;
- **Ethical auditor** — checks for data bias;
- **Regulatory agent** — ensures compliance with standards;
- **Interface designer** — adapts the solution for doctors and patients.

If, say, the task is to optimize a **city's energy system**, the additional agents would include:

- **Power systems engineer** — models load demands;
- **Environmental agent** — assesses the carbon footprint;
- **Economist** — calculates tariffs and subsidies;
- **Crisis manager** — develops contingency plans for emergencies.

Let's return to the step-by-step process of Phase 1.

**PHASE 1 START**

**Step 1.** Initial information gathering (Interviewer agent)

***Actions:***

- asks the user clarifying questions using a template (context, goal, constraints, success criteria);
- records key requirements, expectations and implicit needs;
- identifies contradictions and ambiguities in the request;
- structures the information into a draft problem description.

***Tools***: question checklists, interview templates, active listening techniques.

***Output***: structured draft problem description (in Markdown, Google Docs or plain text file format).

***Result:*** structured set of requirements (e.g., in the form of a table or numbered list).

**Step 2.** Semantic model construction (Modeling agent)

***Actions***:

- creates an ontology of the subject area (entities, relationships, rules);
- defines data types and input/output formats;
- identifies known solution patterns for similar tasks;
- builds a conceptual schema (knowledge graph, UML diagram).

***Tools***: ontology editors (Protégé), graph visualization tools (Graphviz, Neo4j Browser).

***Output:*** semantic task model (in D0SL, OWL, RDF, UML format or as a conceptual schema).

**Step 3.** Formal request generation (Formulator agent)

***Actions:***

- transforms the draft from Step 1 and the semantic model from Step 2 into a clear system request;
- specifies the structure and format of the expected result;
- formulates an LLM prompt or API specification.

***Tools:*** prompt engineering, technical assignment templates.

***Output:*** formalized task (as an LLM prompt, API specification, technical assignment or machine-readable format — JSON/YAML).

**Step 4.** Success criteria definition (Verifier agent)

***Actions:***

- formulates quantitative and qualitative metrics for solution evaluation (accuracy, speed, completeness, compliance with standards);
- describes conditions under which the task is considered completed;
- creates a validation checklist and a set of test cases.

***Tools:*** SLA templates, checklists, testing frameworks.

***Output***: completion criteria (checklist, set of test cases, SLA).

**Step 5**. Data collection and updating (Context aggregator)

Actions:

- searches for relevant external data via RAG, APIs, knowledge bases;
- filters information by reliability and recency;
- creates a "memory" for sequential task solving;
- enriches the context with data from open sources and corporate databases.

***Tools***: RAG systems, API clients, parsers, knowledge bases.

***Output:*** enriched context (set of documents, dataset, source links, structured data).

**Step 6**. Decomposition and planning (Coordinator/Orchestrator)

***Actions:***

- breaks the task into subtasks considering the semantic model and success criteria;
- builds a dependency graph (which subtasks are executed sequentially, which — in parallel; "AND"/"OR" relationships);
- assigns agents to subtasks;
- allocates resources (computational, temporal);

- generates a hierarchical execution plan.

*Tools:* graph building tools, task planners, planning algorithms.

*Output:* hierarchical subtask plan with dependencies and responsible agents (in JSON, YAML format, Gantt chart or task graph).

**Step 7.** Validation and verification (Tester agent)

*Actions:*

- checks the plan's integrity: do all subtasks cover the original goal? Are there any contradictions?
- runs a "dry run" of the plan on synthetic data or a simplified scenario;
- identifies potential risks (data shortage, agent conflicts, bottlenecks);
- makes adjustments to the plan if necessary.

*Tools*: simulators, test environments, validation checklists.

*Output:* approved execution plan with a "ready to launch" mark and a validation report.

**Step 8**. Execution initiation (Coordinator/Orchestrator)

*Actions:*

- launches subtask execution according to the plan;
- transfers control to **Phase 2** (Planning with smart contract generation);
- records initial metrics and monitoring checkpoints.

*Output:* readiness signal for transition to the next phase + start-up report (status "Launch completed", start time, list of launched subtasks).

**END OF PHASE 1**

Key synergy mechanisms in Phase 1:

- **Sequential data transfer** — the result of each step serves as input for the next one (information is neither lost nor duplicated).
- **Agent specialization** — each agent performs a narrow task using its competencies:
    - interviewer — communication;
    - modeler — semantics;
    - verifier — metrics;
    - aggregator — data;
    - coordinator — management;
    - tester — quality control.
- **Centralized management** — the coordinator/orchestrator synchronizes agents' work, builds the dependency graph, and controls transitions between steps.
- **Context enrichment** — the context aggregator updates data at each step, improving the accuracy of formulations and plans.
- **Early validation** — the tester checks the plan before execution, reducing the risk of costly errors during the execution phase.

- **Formalization at all levels** — from a verbal request to a machine-readable plan, ensuring unambiguity and executability of the task.

Let's return to the full lifecycle of task engineering and see how the composition of a Multi-Agent System (MAS) is formed as it progresses. For example, at the stage of transition from the initial need to task formulation, the MAS included **Interviewer agent, Modeling agent, Coordinator/Orchestrator agent** and other. When moving to planning the task solution, the MAS orchestrator may additionally engage the following agents: **Financial agent** (calculates budget and return on investment); **Technical agent** (selects implementation tools); **Operational agent** (designs implementation processes); **Analytical agent** (defines success metrics). During execution of the planned solution, depending on the task being solved, the following agents may appear: **Product agent** (creates digital products); **Marketing agent** (launches advertising campaigns); **HR agent** (organizes staff recruitment and training); **Logistics agent** (optimizes supply chains). Finally, for successful task solution via task engineering, continuous monitoring of the task formulation and solution process and its adaptation to changing conditions is extremely important. For this, the following agents are needed: **Analytical agent** (collects progress data); **Coordinator** (initiates corrective cycles when deviating from KPIs); **Legal agent (**updates documents when regulations and governing documents change).

Generally, when discussing a multi-agent system involved in task engineering, it should be emphasized that its composition is not standard and must be adapted to the complexity, industry affiliation, and other specifics of the need and the task being solved. If necessary, the MAS can be expanded with specialized agents and changed during the solution process.

Summarizing the above, we first of all note the relevance and modernity of task engineering, as it reflects key trends in the development of modern AI technologies:

- **Methodological integrity** — the task-based approach provides a methodological and theoretical foundation for the new discipline, while other elements serve as practical tools;
- **Technological relevance** — the proposed integration aligns with the current trend towards specialization of AI systems (AI agents and MAS), improved interaction (prompt engineering), and knowledge management (task-based approach + context engineering);
- **Development prospects** — the proposed concept of a new scientific-engineering discipline is universal, as it can be easily scaled to new subject areas and thus may become a standard for solving complex, large-scale, interdisciplinary tasks;
- **Practical applicability** — according to the author, task engineering will be especially effective in construction, industry (smart factories), business analytics, logistics (autonomous warehouses), science, medicine (robotic surgeries), public administration, and urban management (smart cities).

The author is confident that the proposed synergistic integration of the task-based approach, prompt and context engineering, AI-agents, and MAS can form the basis of a **universal instrumental-technological task-solving system**, as it allows scalability from individual local tasks of various industry focus to global ecosystems, where: the task-based approach sets the structure and success criteria; prompt engineering ensures precise interaction with models and agents, including through cross-verification of agents;

context engineering helps adapt to environmental changes in real time, thus guaranteeing knowledge relevance; MAS, consisting of software and embodied agents, are capable of implementing solutions to complex tasks in the digital and physical world.

The creation and subsequent development of the platform is conceived by the author as the development and implementation of a set of measures, on one hand, to further improve the methodological and technological foundations of the new scientific-engineering discipline — task engineering, and, on the other hand, to design and subsequently implement the instrumental-technological platform itself. Within and through this platform, in accordance with the methodology and technology of task engineering, specialized AI systems will be created to meet various classes of weakly structured needs (business, finance, medicine, logistics, construction, science, production, public administration, smart cities, etc.).

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
