# OpenReview forum: "TASK-BASED ENGINEERING"
_mathai.club/MathAI/2026/Conference — 2026 Oral_

### Official Review · Reviewer_spA7 · 2026-03-12
**TASK BASED ENGINEERING**

**Rating:** 6
**Confidence:** 3

**Review:**

Summary
This paper proposes the concept of task-based engineering. This unified framework integrates the task-based approach from semantic modelling, modern prompt and context engineering techniques for generative AI, and agent-based architectures. The goal of the framework is to support the transformation of weakly structured needs into well-defined computational tasks and to enable their automated solution using multi-agent systems. The paper presents a detailed conceptual architecture and describes the lifecycle of task formulation, planning, execution, and adaptation.

Strengths:
-Ambitious conceptual synthesis. The paper attempts to unify several important directions in modern AI: semantic modelling, generative AI interaction methods, and agent-based architectures.
Clear motivation. The problem of transforming weakly structured needs into formal tasks is well articulated.
-Connection to a mathematical school. The paper explicitly situates its ideas within the tradition associated with A. N. Kolmogorov, Yu. L. Ershov and S. S. Goncharov, which strengthens its intellectual grounding and links the proposed framework to established lines of research in mathematical logic, computability, and semantic programming.
-Rich contextualization. The work places the proposal within a historical lineage, including semantic programming and task-based approaches developed in logic and AI.
-Potential practical relevance. The proposed framework could be applicable in domains such as smart cities, logistics, or business analytics.

Suggestions for improvement.
The paper could be strengthened by:
-providing a more concise formalization of the proposed framework;
-including illustrative case studies or prototype implementations demonstrating the proposed architecture;
-clarifying which elements of the framework constitute novel methodological contributions compared with existing approaches in agent-based AI and AI orchestration systems.

These additions would make the proposal clearer and help illustrate its practical applicability.

Final Recommendation
POSTED / Poster-style acceptance with minor revision
The paper presents an interesting conceptual integration of several important AI paradigms and may stimulate discussion on the methodological foundations of task-oriented AI systems. With some clarification of contributions and possible illustrative examples, the work could be a valuable addition to the conference discussion.

---

### Official Review · Reviewer_Eeaz · 2026-03-13
**Conceptual discussion but not a technical research**

**Rating:** 2
**Confidence:** 5

**Review:**

Summary:
The paper introduces the idea of task-based engineering as a new discipline that combines the task-based approach, prompt and context engineering, and AI agents.

The paper describes a multi-step workflow where an initial request is analyzed, formalized into a task, planned, and then executed using a coordinated multi-agent system. dinator.

However, the paper is mostly conceptual. It does not contain algorithms, mathematical models, experiments, or empirical validation. The work mainly summarizes existing ideas from prompt engineering, context engineering, and agent architectures and combines them under the concept of task-based engineering

Strengths:

The proposed multi-step lifecycle (need analysis, task formalization, planning, execution, monitoring) is easy to understand and could be useful as a conceptual framework

Weaknesses:
Lack of technical contribution, no experiments or evaluation, no comparison with existing methods, mostly descriptive content

---

### Official Review · Reviewer_kZQn · 2026-03-13
**Methodological proposal lacking mathematical depth and empirical validation**

**Rating:** 4
**Confidence:** 3

**Review:**

This paper proposes "task-based engineering" as a unified discipline integrating three components: the task-based approach (rooted in semantic modeling and Russian mathematical logic traditions), engineering methods for LLM interaction (prompt and context engineering), and agent-based AI architectures. The paper presents a conceptual framework and multi-phase lifecycle for transforming weakly structured needs into formalized tasks solved by multi-agent systems.

**Strengths:**
-  Clear problem motivation as the challenge of transitioning from ill-defined needs to well-formulated tasks is legitimate and important
- Connection to Kolmogorov, Ershov, and Goncharov's work on semantic programming and computability theory provides credible mathematical lineage
- Thorough coverage of relevant background (task-based approach, prompt engineering, MAS architectures)
-  The proposed lifecycle and Phase 1 breakdown offer a clear procedural framework

**Weaknesses:**
- No formal contributions as the paper does not contain theorems, proofs, or formal defintions.
- Paper format is raw, affecting negatively to readability (no section division, too much listing)
- Some citations regarding mathematical foundations are mentioned without integration or presentation.
- Lack of complexity analysis (formal guarantees about convergence, completeness, or computational bounds)
- Lack of empirical evaluation (no experiments, case studies, benchmarks, or comparisons with existing MAS frameworks
- Some claims like "synergistic integration" lack theoretical and empirical support
- Breadth of claims is excessive and lack domain-specific considerations.

**Assesment:**
The paper combines existing paradigms (semantic modeling, prompt engineering, MAS orchestration) but doesn't demonstrate that this integration is novel or that existing systems don't already do this implicitly. The most of the paper establishes reference concepts while lacking the original experiments. This is fundamentally a systems/methodology paper, not a mathematical contribution. For acceptance, the authors would need to:
1. Provide formal mathematical analysis of task decomposition and agent coordination
2. Prove properties of the proposed framework
3. Demonstrate empirical validation on concrete problems, and
4. Narrow scope to enable genuine depth.
5. Improve the paper structure by converting existing lists to algorithmic format and assigning proper section divisions.

---

### Decision · Program_Chairs · 2026-03-14

**Decision:**

Accept (Oral)

**Comment:**

Dear Author(s),

On behalf of the Program Committee of the International Conference on Mathematics of Artificial Intelligence (MathAI 2026), we are pleased to inform you that your paper has been accepted for an oral presentation at MathAI 2026.

Your paper was evaluated through a rigorous two-stage review process involving both automated screening and expert review by members of the Program Committee. The reviewers recognized the quality and contribution of your work.

Presentation details:

- Format: Oral presentation (15–20 minutes + 5 minutes Q&A)
- Mode: You may present either in person (offline) at the conference venue in Sirius, Russia, or remotely via Zoom. Please indicate your preferred mode when confirming your participation.
- Conference dates: Marh 30 - April 3, 2026
- Website: https://mathai.club

Next steps:

1. Please confirm your participation and presentation mode by replying to this email mathai.club@yandex.ru no later than March 15, 2026 18:00 Moscow time.
2. If you plan to attend in person, the organizing committee will provide accommodation details separately.
3. Please prepare your final camera-ready manuscript according to the formatting guidelines available at https://mathai.club and upload it to OpenReview by March 15, 2026 18:00 Moscow time.

Should you have any questions regarding the program, logistics, or your presentation slot, please do not hesitate to contact us.

We look forward to your contribution to MathAI 2026.

With kind regards,

MathAI 2026 Program Committee
International Conference on Mathematics of Artificial Intelligence
https://mathai.club
OpenReview: https://openreview.net/group?id=mathai.club/MathAI/2026/Conference
Telegram: https://t.me/MathAI_club
Email: mathai.club@yandex.ru